# Dynamic Information Flow Tracking: Taxonomy, Challenges, and Opportunities

**DOI:** 10.3390/mi12080898

**Published:** 2021-07-29

**Authors:** Kejun Chen, Xiaolong Guo, Qingxu Deng, Yier Jin

**Affiliations:** 1School of Computer Science and Engineering, Northeastern University, Shenyang 110016, China; kejunchen@ufl.edu (K.C.); dengqx@mail.neu.edu.cn (Q.D.); 2Department of Electrical and Computer Engineering, Kansas State University, Manhattan, KS 66506, USA; guoxiaolong@ksu.edu

**Keywords:** dynamic information flow tracking, data-flow integrity, control-flow integrity, data-flow analysis

## Abstract

Dynamic information flow tracking (DIFT) has been proven an effective technique to track data usage; prevent control data attacks and non-control data attacks at runtime; and analyze program performance. Therefore, a series of DIFT techniques have been developed recently. In this paper, we summarize the current DIFT solutions and analyze the features and limitations of these solutions. Based on the analysis, we classify the existing solutions into three categories, i.e., software, hardware, software and hardware co-design. We discuss the DIFT design from the perspective of whole system and point out the limitations of current DIFT frameworks. Potential enhancements to these solutions are also presented. Furthermore, we present suggestions about the possible future direction of DIFT solutions so that DIFT can help improve security levels.

## 1. Introduction

Dynamic information flow tracking (DIFT) is a technique which leverages metadata tags to track the information flow among different entities. DIFT can be used to provide runtime protection for programs; data usage analysis; and performance analysis, wherein the tag is used to record the security properties of data and instructions. The program and its data usage profile can be acquired through analyzing and checking the tag usage. While the implementations of DIFT vary based on the target designs and other constraints, we can roughly divide DIFT techniques into three types: (i) software-based DIFT, (ii) hardware-based DIFT and (iii) software and hardware co-design-based DIFT.

The different types of DIFT each have their benefits and limitations. For software-based DIFT mechanisms [1,2,3,4,5,6,7,8,9,10,11,12,13,14], the tag can be combined with the software context closely through binary code instrumentation and source code modifications. Since one does not have to modify the hardware components, the software-based DIFT design provides better flexibility, customization and scalability. However, software solutions often introduce high performance overheads because additional instructions are added to support DIFT. Meanwhile, hardware assisted DIFT design [15,16,17,18,19,20,21,22,23,24,25,26,27,28,29,30] can extract sufficient runtime data from the hardware level to enforce different security rules. By modifying the hardware, all DIFT-related operations are implemented as hardware logic to accelerate the tag propagation and tag checking. An exception is raised once the security rules are violated. This type of design reduces the high performance overhead but at its own cost. Hence, it is difficult to deploy hardware assisted DIFT in a real system, especially in modern commercial systems. The hardware-based DIFT design may not provide much flexibility and scalability after deployed. The hardware overhead usually cannot be handled by resource-constrained devices, e.g., low-end embedded systems.

In addition to the above DIFT design, the hardware and software co-design [31,32,33,34,35] is a good option for enforcing DIFT rules. This type of design can extract information from a special hardware interface and associate the software context with the data from hardware. Flexible and fine-grained security check rules can be enforced in this type of design. There are some open questions. For example, it is difficult to decide on how to design a suitable tag when considering both software and hardware. Security rules on the software side need to be translated into tags. The hardware needs to be extended or modified to support the tags. Additionally, tag design needs to support rule updating after the system is deployed. It is difficult to balance both flexibility and hardware overheads at the same time.

The above discussion briefly compared the different DIFT types. Additionally, there are some surveys about existing DIFT works. Berner et al. in [36] summarized and analyzed several DIFT works. Ormeir et al. in [37] aimed at analyzing the existing methods of malware detection and analysis. Additionally, this work does not involve static methods. Yan et al. in [38] mainly focused on the DIFT works on mobile malware detection. They did not give a comprehensive analysis on existing DIFT works on different application environments. Yan et al. in [39] analyzed the precision and soundness of existing software-based DIFT works. D’Elia et al. in [40] analyzed existing dynamic binary instrumentation (DBI) methods. Additionally, they discussed the usage of DBI in the DIFT framework. Still, a more detailed comparison between different DIFT solutions is needed to guide the use of these solutions. Thus, in this paper, we analyze and systematize all three types of DIFT design—hardware-based, software-based and software and hardware co-design. The implementations of these three types of DIFT design cover different computer system layers. Specifically, for hardware-based designs, we mainly discuss the gate-level-based DIFT design and micro-architecture-level-based DIFT design. For software-based solutions, we focus on information flow tracking techniques targeting the software and the operating system. For software and hardware co-designs, we cover a variety of DIFT solutions which use both software and hardware to enforce information flow tracking. Table 1 shows a high-level comparison among all DIFT solutions mentioned herein. Based on the analysis, we further discuss the possible solutions to help overcome these limitations. We also discuss potential research directions for better DIFT techniques.

The main contributions of the paper are summarized as follows.

We summarize a variety of existing DIFT solutions and classify them according to their designs and implementations as hardware-based, software-based or software and hardware co-designed.We analyze the advantages and limitations of different DIFT solutions. Based on the analysis, we present a comparison between different DIFT implementations.We discuss possible solutions to the limitations of the current work. New research directions for DIFT techniques are also presented.

## 2. Background

In this section, we introduce the background and basic knowledge about the DIFT framework regarding tag setup, propagation and checking.

### 2.1. Tag Size and Granularity

The size and format of a tag decides the maximum amount of security rules which can be supported. As shown in Table 1, different types of DIFT designs have different tag structures. For hardware-based gate-level and register transfer level (RTL) DIFT designs [15,16], logic gates and registers are extended to support the extra tag. For a gate-level-based DIFT design, an extra tag tied to a logic gate is added according to the original gate logic. The information flow in software running on such hardware logic will be tracked. The customization of security policy cannot combine the software context with the gate logic directly. As a result, false positives may be caused. For example, the benign data will be tainted following an *OR* tag propagation rule. Further, the security check is significantly influenced by the tag computing rules. For micro-architecture-level DIFT design [21,22], the basic granularity of the tag depends on the data width of processor. There are two common granularities, word-level tags and byte-level tags. In a simple implementation, the tags represent two basic states of data, i.e., trusted and untrusted. In this case, it is difficult to differentiate the source of untrusted data. Moreover, this type of tag cannot be used to represent multiple-level security policies. To overcome this limitation, a simple method is to adopt more fine-grained and larger tag sizes such that more security policies can be supported. However, a larger tag size may cause a waste of tag storage, since not all data are involved in computing. Many tag bits may not be used at runtime. In addition to representing the states of data, the tag can also be used to indicate different security policies with different instruction types [21]. Different tag propagation and tag check rules are customized according to corresponding instruction types.

For software-based DIFT designs, the tag design is more flexible than for their hardware-based counterparts. The authors of [10] designed a variable-length tag structure to support flexible and programmable security policies. The authors of [1] assigned tags to different objects in the operating system instead of data. However, a flexible tag structure introduces large performance overheads and increases the complexity of tag storage.

### 2.2. Tag Storage

In gate-level DIFT design, the tag is stored in an extra gate logic. For micro-architecture-level DIFT design [21,22,28], the tag is often stored in registers or memory of a modern processor. Therefore, the register files and memory need to be extended to support tags. In addition, the processor pipeline stages need to be modified to transfer the tag in parallel with the instructions bein executed. There are two basic types of tag storage, normal storage and shadow storage. For normal storage, the register files need to be extended to support the corresponding tags. Thus, the normal memory can be partitioned into normal data memory and tag memory. Extra access controls on tag memory should be enforced to prevent illegal tag modifications. In shadow storage, extra storage are allocated to store tags. Normal instructions cannot access the storage, as the shadow storage is not connected to the data bus. However, this type of storage is difficult to expand, especially after deployment. For system-on-chip (SoC)-level DIFT design, tag storage can be allocated to either IP wrappers or other dedicated hardware components and is often customized based on the target IP.

For software-based DIFT methods, the tag is stored in the same storage space as other data. For instance, the DIFT solution proposed in [11] stores the tag adjacent to data for spatial locality. This may cause large performance and storage overheads, as the tag fetching requires extra clock cycles for memory access. Even adopting byte-level tags introduces a 25% storage overhead on a 32-bit processor. In addition, extra security checks need to be performed on tag storage to prevent illegal tag accesses and modifications.

For a software and hardware co-design-based DIFT framework, the tag storage can use normal storage or shadow storage. New instructions may be inserted to support the tag management [33] and to access the shadow storage [35].

### 2.3. Tag Propagation

One fundamental and important operation in DIFT is the propagation of the tag from source to destination. In the hardware-based DIFT, the tag propagation can be divided into two types: (i) the tag propagation operation accompanied with instructions in the pipeline [21,22]; and (ii) dedicated hardware logic for tag propagation, e.g., coprocessors [24,25,27,28]. Within the scope of hardware-based DIFT, in gate-level DIFT methods, the tag is propagated in parallel with the data flow through gate logic. For micro-architecture-level DIFT design, the tag is propagated between registers and memory. For SoC-level DIFT design, the tag is propagated through the bus in parallel with the corresponding bus request or data transfer. In order to support the tag propagation, the bus needs to be extended. Alternatively, the tag may be transmitted at the bus transaction initialization phase.

In software-based DIFT design, the information related to the target program is extracted using binary instrumentation or static analysis. Then the DIFT-related instructions are inserted to propagate the tag from one entity to another entity. The granularity of an entity depends on the implementation details, e.g., variables, basic blocks and threads [2]. Tag propagation in software and hardware co-design needs the cooperation between software and hardware. Extra DIFT-related ISA [31] is introduced to perform the tag propagation in the hardware. Additionally, the security policy can be defined using software and the tag propagation is automatically executed at the hardware layer according to the defined security policy.

For software and hardware co-design-based DIFT, the tag is often implemented at the hardware level. The DIFT operations are inserted into the target program to instruct the processes of tag propagation and tag checking. The authors in [31] used binary instrumentation to convert all implicit flows into explicit flows. Every instruction in standard ISA is assigned corresponding IFS ISA to instruct the tag-related operations at the micro-architecture level. In addition to the extended ISA, the authors in [41] modified the kernel to support a timestamp which defines the life cycles of corresponding data. Furthermore, the tag at the micro-architecture level will be propagated according to the timestamp defined at the kernel level. In [33], DIFT operations are converted to symbolic values instead of actual instructions. Processor pipelines will be extended to support the symbolic execution and corresponding tag propagation.

### 2.4. Tag Checking

The security rules are enforced at the tag checking stage. In hardware-based DIFT designs, the tag checking operation is implemented using dedicated hardware logic. The tag checking rules are configured through either global registers [21] or custom instructions [27]. The software-based methods execute tag checking combined with software context closely. Note that the tag checking uses extra instructions and will cause extra clock cycles. The hardware tag and software context are combined closely in co-design-based DIFT. The security policy in the software part can be expressed as the combination of tag operations in the hardware layer.

The handling of the exception is another essential part in the tag checking stage. In software-based methods, an exception is raised through a software interrupt in the target program that monitors the program in other threads [2]. The hardware-based DIFT design may add a new hardware interrupt or extend the current hardware exception signal to support the exception handling when executing DIFT operations. However, the normal interrupt and exception handling may engross the whole OS due to the high performance overhead. In order to reduce the overhead caused by exception, the authors in [21] adopted the user-level exception and extra machine mode. In software and hardware co-design-based DIFT, a hardware exception is raised in the hardware layer when the tag checking fails. This exception will be handled at the software level with the intervention from the user or the developer.

## 3. Hardware-Assisted Dynamic Information Flow Tracking

In this section, we discuss the hardware-based DIFT designs, including gate-level DIFT designs and micro-architecture-level DIFT designs. The gate-level DIFT designs include gate-level netlist and RTL designs. For micro-architecture-level design, we mainly focus on in-core DIFT (the tag-related operations are integrated into the processor pipeline) and off-core DIFT (the tag-related operations are packaged at the back end of the commit stage in the processor pipeline). We discuss the offload DIFT, wherein target applications and DIFT applications run on different processor cores, in Section 4 and Section 5.

### 3.1. Gate-Level DIFT

Tiwari et al. in [15] presented a DFIT scheme on gate-level netlist to track all information flows. This design resolves the problem of implicit flow and coverts flow to some extent because all tracking logic is implemented as gate logic. The tracking gate logic can reflect how untrusted data influence trusted data in different branches. Both data and code can be tracked because all information flow is visible from the viewpoint of the hardware. The tracking operations run in parallel with the normal operations. The information flow checking operations are combined with the basic ISA. Therefore, the software does not need to be instrumented or modified. The gate-level DIFT can be easily configured to enforce different security rules. However, extra tracking operations introduce high hardware overheads.

Tiwari et al. in [16] presented an architectural framework to support verifiable information flow tracking. The whole system is partitioned into a trusted part and an untrusted part. Based on the partition, the time needs to be multiplexed between the trusted part and the untrusted part to prevent the sensitive information leakage. In order to support the above functionality, this framework includes a verifiable critical function in gate-level implementations, e.g., pipeline, cache and I/O systems. Moreover, the microkernel contains a scheduler to support the context switch between trusted and untrusted partitions.

Ardeshiricham et al. in [17] presented an information flow tracking (IFT) method to support the verification of security properties in RTL code. This work provides two tracking libraries with different precision levels to generate information flow tracking logic. However, the IFT logic introduces false positives/negatives due to the tag propagation rule, e.g., through the *OR* rule and the *AND* rule. For example, the *OR* rule introduces a high false positive value because the security tag is set whenever one of the inputs is set.

Tiwari et al. in [18] presented an architecture to enforce non-interference between trusted entities and untrusted entities and allow the related security properties be verified at gate level. They designed an execution lease between a caller (leaser) and a callee (leasee). The control flow is bounded between two contexts using a timer. The program counter is reset to restore PC once the timer expires. Additionally, the address space is bounded by using extended instructions to enforce access control on different parties.

### 3.2. Micro-Architecture-Level DIFT Design

#### 3.2.1. Dynamic Information Flow Tracking on SoC

Piccolboni et al. in [19] proposed an IP wrapper to protect access to IPs from software. The DIFT operations are implemented as hardware logic in the IP wrapper to check for malicious access. The tag is issued with accompanying with the memory access. They provide three types of DIFT logic in the IP wrapper: (i) configuration logic for checking the data in the configuration register; (ii) load logic for checking the input data; and (iii) storage logic for generating the tag for output data. Porquet et al. in [20] enforced DIFT operation on a bus and IP through a whole design. In this work, every IP was assigned a wrapper to receive the tag from the bus. They extended the bus interconnect to acquire the tag from normal memory. The data and tag were stored separately in normal memory. A two-level table for tag management was adopted. The first-level tag was used to indicate whether the current page was tagged. The second level tag indicated word-granularity in page.

#### 3.2.2. In-Core DIFT Design

Dalton et al. in [21] presented a DIFT architecture to support a flexible security configuration at runtime. Differently from common DIFT design and the tag-based architecture, the tag in [21] was used to represent the security policy for propagation and check instead for data states, e.g., trusted vs. untrusted. In order to support flexible software programming, the authors provided two global configuration registers, i.e., the tag propagation register (TPR) and the tag check register (TCR), to configure the security policy at runtime. The configuration register could be configured only in trusted mode. Moreover, the tag propagation and check could only be disabled in trusted mode. Further, user-level exceptions were used to reduce the overhead of crashing the whole OS. The OS switches to trusted mode but keeps the address space once the DIFT exception is raised. However, the security policy is difficult to update when the architecture is deployed. The processor pipeline, register and memory system need to be extended to support DIFT operations. Similarly to [21], Palmiero et al. in [22] also adopted global configuration registers to customize the rule of tag propagation and checking. Note that the rule customization depends on the instruction type. Different types of instructions have different rules. Their method provides more fine-grained tracking than [21]. However, it lacks enough flexibility for security policy reconfiguration for different program contexts. The DIFT schemes proposed in [21,22] need to extend or modify the original pipeline stages to support the DIFT-related operations. For the same reason, this type of DIFT design is difficult to be deployed on modern commercial processors. The scalability of customization of security policies is also a challenge for this type of DIFT design.

Differently to above in-core based design, Li et al. in [23] simplified the common DIFT framework to only track registers. The states of registers are analyzed to determine whether return oriented programming (ROP) attacks happened. In order to reduce the modifications to existing processor architecture, the taint related information is stored in the shadow renaming tables and reorder buffer.

#### 3.2.3. Off-Core DIFT Design

Lee et al. in [24,25] used the core debug interface (CDI) to extract a trace of a program at runtime from a processor core. The CDI is mainly used as a special interface for functional verification and program performance analysis. This interface is normally connected to on-chip debuggers which allow the programmer to analyze the contents and states in the register memory. Based on this dedicated component, this work proposes a DIFT engine as a hardware IP attached to a shared interconnector. The runtime trace is extracted from the processor and sent to the DIFT engine for further processing. A heuristic analysis method is applied to decide whether control-flow integrity and data-flow integrity are violated. Similarly, Wahab et al. in [26] used the ARM CoreSight debug component to extract the trace. However, the commercial core debug component could only extract limited information about the program executing on the processor. As a result, the target programs needed to be instrumented to recover the complete program trace. To achieve this goal, tag dependency instructions were generated through static analysis to instruct the corresponding tag operations, e.g., tag propagation and tag checking. Note that extra storage needs to be added to save the tag dependency instructions.

Venkataramani et al. in [27] used an dedicated coprocessor to perform DIFT operations. This type of design does not involve many processor pipeline modifications, which are often complex tasks. Instead, the processor pipeline only needs to be extended slightly to output the committed instructions from the commit stage of pipeline. Additionally, this design decouples the tag and data. It uses a dedicated L1 cache as tag storage to reduce the hardware overhead. Custom instructions are also added to support flexible configurations that adapt to security rules at runtime. However, the design also brings in a new problem, i.e., inconsistency between data and tags on multiprocessors. Each processor is assigned one accelerator to support DIFT operations and it is difficult to manage multiple accelerators on chip multiprocessor (CMP) platforms. Following in the work in [27], Kannan et al. in [28] designed a coprocessor interface which is used to collect the program profiling and machine states from the main processor, e.g., program counter value, memory address and instruction encoding. After receiving the above runtime information, the four-stage pipeline coprocessor analyzes the committed instructions and executes the corresponding tag-related operations. The tag is saved in the dedicated tag cache of the coprocessor to help accelerate the process of tag fetch.

### 3.3. Tagged Memory Architecture

Robert et al. in [29] proposed a complete tagged memory architecture to support information flow tracking and security analysis. A single-bit tag shadow space is used to record the metadata at runtime. An in-DRAM tag table and tag cache are utilized to manage the tag in the whole system. Weiser et al. in [30] presented a new tagged memory architecture to combine compartmentalization (e.g., Intel SGX [42]) and isolated execution environments (e.g., TrustZone [43]). Two extra tag bits are utilized to differentiate privilege levels and security domains. A dedicated piece of hardware serves as the trust root to enforce security policies in terms of tag and finish security domain switching. Additionally, extra tag-aware instructions are added to support flexible tag management.

## 4. Software-Based Dynamic Information Flow Tracking

In this section, we will introduce common software-based DIFT designs. We classify the software-based DIFT designs into two types: (i) system-level DIFT design; and (ii) program level DIFT design. For system-level DIFT design, the security analysis are operated on real scenarios, e.g., Operating System and CMP platforms. Furthermore, the inconsistencies between the original data and metadata (tag) need to be solved efficiently. The program-level DIFT design uses target software through binary instrumentation and source code modifications to support DIFT operations.

### 4.1. System-Level Dynamic Information Flow Tracking

Zeldovich et al. in [1] presented an operating system supporting information flow tracking. They tried to accomplish the following two goals: (i) reducing the amount of code that needs to be trusted; and (ii) separating the trusted function from untrusted functions. The tag design of their work is different from the common DIFT design, as their method assigns a tag for the basic kernel object type, e.g., thread, address space, segments, gates, containers and devices. The tag is checked and accompanied with the information flow between different objects. A *container* is used to enforce access control and tag allocation on different objects. This work combines the information flowing tracking with operating system abstraction effectively. However, a new programming model is required to enforce security rules on target programs.

Nagarajan et al. in [2] presented a new infrastructure for DIFT operations on CMPs. Supported by this infrastructure, the DIFT operations are performed in a helper thread and the target program runs on the main thread. This work uses a dynamic binary translator (DBT) to manage the helper thread and the main thread. The DBT is responsible for generating additional code in the main thread, and the helper thread is used for the communication between these two threads. The main thread communicates with the helper thread through the shared memory. Exceptions and interrupts are raised in the helper thread once any security rules are violated. This design introduces higher performance overhead for two reasons: an inter-thread communication overhead and the instructions required for DIFT operation. The helper thread needs to follow the main thread’s control flow and only execute the DIFT-related instructions.

Differently from the sequential DIFT tools in [2], Ruwase et al. in [3] presented a parallelized DIFT framework to speed up the tag checking on CMPs. They adopted multiple worker (or helper) threads to check the tag in order to reduce the performance overhead. However, the serial dependencies need to be solved before executing DIFT operations. Their method tracks the tag symbolically rather than tracking the tag values. The tag symbols are saved temporarily in the symbolic inheritance table until the tag value can be determined explicitly. The binary operation presents a new challenge because each node relies on the tag value from two nodes. To solve the limitation, leveraging the log based architecture (LBA), a master thread is introduced to assign the task to worker threads and wait for the summary from worker threads. Additionally, every worker thread is responsible for the unary node.

In addition to the software solution to solve inconsistency on data and its metadata, Kannan et al. in [4] used hardware to track the data and enforce the same ordering on metadata. Every instruction is assigned an ID shared between the main thread and the worker (or helper) thread. According to the ID, worker threads deal with the requests and follow the instruction order. Moreover, Nightingale et al. in [5] decoupled the program and replayed the program in other cores. They used a function *fork* to copy the runtime states of the program from the target core to other cores when a security check was executed. The target program can then be continued on other cores. Next, the security check operation was executed in the manner of speculative execution. The affected states can be rolled back once the security check fails. However, this method may introduce other speculation based attacks such as the Spectre [44].

Arefi et al. in [6] proposed a whole system-level DIFT framework to resist in-memory injection attacks, which are difficult to be detected using current malware analysis solutions. In order to capture the indirect flow in security applications, they introduced different types of tag instead for expressing different security policies in terms of similar tag structure. Besides, tags convey rich provenance information, e.g., the lifetime of a data byte and activities associated with the data byte. Further, indirect flows can be processed in different security policies according to the tag type. Similarly to other system-level DIFT solutions, significant performance and memory overheads are incurred (56x performance slowdown was introduced in a QEMU-based simulator platform).

### 4.2. Program-Level Dynamic Information Flow Tracking

Cheng et al. in [7] proposed an instrumentation method to track the tag for the target program and shared libraries dynamically. This system includes four parts: a configuration file, shadow memory, a program monitor and a loader. The configuration file is responsible for security policy configuration. Shadow memory is a special structure to hold the tag. The program and its metadata are loaded at the initial stage using the loader. The program monitor activates the customized security policies and raises an exception when the tag check fails. This type of method may incur a high performance overhead for two reasons: an instrumentation overhead and a runtime overhead. The influence of the instrumentation overhead is relatively small because of modern storage capabilities. The runtime overhead comes from register spilling, tag mapping and tag propagation. In order to further reduce the whole performance overhead, the runtime overhead needs to be optimized. Chen et al. in [8] presented an efficient static information flow tracking method in order to reduce the performance overhead. Instead of using DBI-based methods, they adopted static binary instrumentation (SBI) to scan sources of taint and select the instructions need to be implemented and monitored. They proposed an approach to statically identify the instructions that will involve tainted memory or registers through value set analysis (VSA).

Similarly, Castro et al. in [9] presented a basic software framework to enforce data-flow integrity. This framework includes two basic steps: i) using static analysis (reaching definition analysis) to generate a data-flow graph (DFG); ii) running the target program to ensure the data flow follows the DFG at runtime. This method introduces a high performance overhead and many false negatives because of the more conservative static analysis.

To support more flexible and programmable dynamic taint analysis, Clause et al. in [10] presented a framework to perform data-flow and control-flow taint analysis. This framework provides variable tags for different types of program data, including variables, memory locations, function calls and I/O streams. Therefore, a software programmer can specify different tags for different data to perform various analyses, e.g., program debugging, program testing and vulnerability detection. Further, the data can be assigned multiple tags at the same time for different usages. The tag checking and propagation rules can be configured to examine different tags. The flexible tag design introduces a high performance overhead caused by programmable tag checking and propagation rules at runtime, and a storage overhead. Enck et al. in [11] presented an IFT system for smartphones to provide adequate control over user’s privacy information. The main goal of this system is to monitor the usage of sensitive user information on third-party applications. Therefore, multiple sources of sensitive data need to be tracked at runtime. This IFT system provides four basic granularities of data tracking, i.e., variable-level, method-level, message-level and file-level, to enforce complete data tracking on smartphones. For variable-level tracking, every variable is assigned a bit vector to hold 32 different taint markings. The tags are stored adjacent to variables to provide spatial locality. Additionally, the size of stack frame allocated is doubled. The message-level tracking is responsible for the messages between different processors. The method-level tracking is used for system-provided native libraries. The consistency of taint metadata is ensured by file-level tracking. The tag propagation logic for object reference, native code, IPC and secondary storage is implemented in the virtual machine (VM) interpreter.

The authors in [12] aimed to provide a shared dynamic flow tracking (DFT) library for commodity software without modifications. This framework uses three types of locations, including instructions, function calls and system calls. The user defined callbacks will be injected into the target program to instruct the tag propagation and checking. The injected code is translated via just-in-time (JIT) compiler to ensure that the injected code runs.

Ji et al. in [13] used record–replay technology to develop a log based system to record the system call events and related activities. Based on the log collected at runtime, a provenance graph is constructed to improve the system performance. The unrelated instructions and processes are filtered out using reachability analysis on the provenance graph. However, the memory overhead introduced is an additional 50%.

In addition to software-based DIFT design, Ferraiuolo et al. in [14] verified practical security architecture through static information flow analysis. The target security architecture can be verified with information flow control (IFC) at the level of the hardware description language (HDL). The main goal of IFC HDLs is to guarantee the security property noninterference at the design stage.

## 5. Software and Hardware Co-Design-Based Dynamic Information Flow Tracking

In this section, we will discuss the hardware and software co-design DIFT framework. This type of design combines the features of both software DIFT and hardware DIFT. Using the binary instrumentation and modified compiler, the hardware and software co-design DIFT framework can provide flexible security rule configuration and fine-grained protection. Meanwhile, the dedicated hardware logic is used to reduce the software performance overhead significantly. Furthermore, the target program can be protected at runtime through cooperation between software and hardware.

Vachharajani et al. in [31] used binary instrumentation and architectural support to enforce information flow security policies at runtime. The conventional ISA is translated into information-flow security (IFS) ISA through the dedicated binary translator. Every instruction in the standard ISA has a corresponding instruction in IFS ISA. Every instruction in IFS ISA is assigned extra security registers to hold the tag. The IFS ISA space is divided into four basic types, i.e., ALU, store, load and branch. The semantic of the standard ISA does not change. Additionally, the information flow check is performed in parallel with the standard instruction. For implicit flow, the binary translator appends the security registers of branches to the possible operands of all instructions that may use the values influenced by the branch.

Santos et al. in [32] presented a new infrastructure to solve the coherence issue between normal data and metadata processing. The processor core, coprocessor and memory hierarchy are modified in this new infrastructure to enforce metadata coherence. A coherence unit is added to track all data access events on the shared bus from the processor core. Furthermore, the access from multi-cores is synchronized in the coherence unit. However, this design introduces high complexity compared to the hardware DIFT design.

Dhawan et al. in [33] proposed a flexible and programmable tag-based architecture model for security policy customization. The security policy is defined as a function mapping between the input set and output set under a specified operation. The security policy is enforced at the level of instructions to enforce memory safety, control-flow integrity, data flow integrity and separation between code and data. Meanwhile, the enforcement of security policy includes the collaboration between symbolic rules in the software layer and concrete rules in the hardware layer. Furthermore, the security policy customized by software developers can express the security policy using abstract symbolic tags. In addition, this architecture supports multiple orthogonal policies at the same time because the rules can be achieved by pointers to the tag tuples.

Townley et al. in [34] analyzed the temporal locations of DIFT operations on tagged data in regard to two features: (i) the number of instructions manipulating tagged data and (ii) the duration of taint-free epochs. They found the distribution of tag access across memory and the accuracy of coarse-grained tags may influence the DIFT design. Based on these observations, they used the feature of strong temporal locality to design a lightweight, coarse-grained DIFT hardware module. The dedicated DIFT module receives the committed instructions from the pipeline. Then, the DIFT-related operations are finished in the dedicated hardware module. The dedicated hardware module can be combined with software DIFT on the single-core and multi-core platform to significantly reduce the performance overhead. The dedicated hardware module will also reduce the hardware overhead. However, it is difficult to apply this framework to general computing platforms. The different access behavior of the program may introduce false positive rates. Additionally, coarse-grained designs may not provide enough protection for the target program.

Liu et al. in [35] enforced data-flow integrity through an open-source tagged memory architecture based on RISC-V SoC. This work included a hardware extension of RISC-V SoC and static analysis to generate CFG. They combined the in-core design and off-core design to finish tag propagation and tag checking at runtime. Through static analysis, the extended instructions were executed to indicate how to propagate tag between memory locations and registers. Additionally, the custom coprocessor instructions are inserted to inform the coprocessor to check the specified tags.

## 6. Discussion and Prospect

In this section, we discuss the benefits and limitations of existing DIFT works. We also present some promising future directions for more efficient DIFT frameworks.

### 6.1. DIFT Works on Different Layers of Whole Systems

As shown in Figure 1, we give an overview of DIFT techniques on different layers of whole systems. In the *program-layer*, the target program is modified further to support the DIFT-related operations through binary instrumentation and compiler aided methods. In the *OS layer*, the target program and DIFT operations are divided and placed into different threads (main thread and helper threads) on different processor cores. The synchronization between the main thread and helper threads is finished through shared memory and a special hardware channel, e.g., a FIFO message queue. The hardware-assisted methods provided in the *architecture layer* help accelerate the DIFT-related operations. The corresponding hardware components and modules are extended to support the tag-related operations.

Full-system-based design is a good candidate for DIFT frameworks because this type of design acquires the essential information from the three layers mentioned above and combines the benefits from both software and hardware DIFT designs. In addition to the tag-related logic in the *architecture layer*, this layer also needs to provide isolation an environment for secure execution. The storage of data and corresponding tag need to be protected properly. Using a hypervisor may be a good method to protect the tag storage and DIFT-related handler. Based on the architectural support, the OS enforces the security rules and provides a mechanism to ensure the security features provided from the *architecture layer* can be used in a safe manner. For example, it is possible to implement an extra machine mode, e.g., monitor mode, orthogonal to the user mode and supervisor mode. Further, the DIFT-related operations can only be finished in this monitor mode. Finally, the program in the *program layer* is used to run the instructions, execute the tag-related operations and invoke the extra machine mode provided by the OS.

### 6.2. Overtaint on DIFT

The problem of overtaint refers to the tag polluting other data when the tag is propagating at runtime without being restricted. Furthermore, it is the main cause of high false positive rates. For example, the tag is propagated to the general purpose register through ALU instructions. The data in the memory location are also tagged when executing memory instructions, e.g., store operations. Similarly, the data in the general purpose register are also polluted by tagged data from memory. In this case, the benign data are also tagged once they are involved in DIFT operations. As a result, the exception is raised even if there are no actual attacks or security risks. Even worse, multiple processors’ data in CMPs platform are polluted because they share the same data sources, e.g., memory or a cache data page.

A better solution to this problem is to restrict the valid region for a tag. The authors in [18] used a watchdog timer to ensure that the control flow is bounded between trusted entities and untrusted entities. However, this type of design is not suitable for tag-based systems because you cannot make sure the exact time bounds for different system events, e.g., system calls, exception handling and memory access. The target program needs to be analyzed and be divided into entities of different granularities, e.g., task, function, basic block and variable. Furthermore, different tags are assigned to different entities to indicate their granularity. In addition to recording the entity type, the tag also needs to record the data source, e.g., using a unique identifier in a limited scope to represent a different source. The tag design also needs to be combined with the actual software context. For example, the security-critical function is treated especially and using a special tag.

### 6.3. Implicit Flow and Explicit Flow

The problem of implicit flow is an open problem in DIFT design. The authors in [31] pointed it out that identifying all implicit flows is a very complex problem. As shown in Table 2, the explicit information flow cannot track nor determine whether the variable *y* is influenced by the branch statement when *x* is not equal to *true*. In order to determine all implicit flows, the statements influenced by control-flow related instructions need to be identified. The common methods for dealing with implicit flow include: (i) static analysis in [45] to find all implicit flows and assign tag operations to the corresponding instructions and data; and (ii) transforming the implicit flow into explicit flow through the static and dynamic method in [31]. However, the above methods work at the cost of performance overheads and precision. Therefore, how to identify all implicit flows and track the implicit flow at a reasonable cost is still a challenge for DIFT designs.

**Listing 1 micromachines-12-00898-t002:** An example of implicit flow.

if (x == true)
y = true;
else
y = false;

### 6.4. A Flexible Tag-Based System

For in-core DIFT design, the adopted global security policy configuration registers provide flexible customization for security policies to some extent. However, the global configuration register is not suitable for the actual software context. The security policy needs to be configured frequently at runtime to adapt to different program environments. In some cases, this type of design may incur high false positive rates. That is, the security policy customized by global configuration registers cannot provide fine-grained protection which is suitable for the actual context.

Therefore, fine-grained tags are designed to express the atomic tag operation, e.g., the tag bit operation, tag fetch and update. The security policy can be expressed using a combination of multiple atomic tag operations. In addition to the flexible expression of tags, the variable length of a tag is considered to support multiple security policies at one time. At the hardware layer, the tag-related operations are executed in extended processor pipeline stages or added coprocessor. For the extended processor pipeline, extra tag-related instructions are executed to finish tag-related operations. Therefore, the extra software performance overhead is introduced. A better solution to this problem is to adopt parallel hardware pipeline stages to process the tag-related instructions instead of extending the original pipeline stages. In the coprocessor, the extra tag-related instructions are counted as nor-operation instructions in the processor pipeline. However, this also influences the performance overhead. A possible solution is to divide the normal instructions and tag-related instructions and place them in different memory locations. Then, the coprocessor can receive the committed instructions from the processor and fetch the tag-related instructions according to the committed instructions.

### 6.5. SoC-Level DIFT Design

Most DIFT designs focus on protecting the target software running on the processor. Only a few methods have been proposed for SoC-level DIFT designs to protect the information flow between third party IPs (3PIPs). The current DIFT design can provide protection for information flow between the processor and 3PIPs because most IPs can be accessed in the form of memory mapped I/O (MMIO) registers. The security check can be enforced on the corresponding memory regions. However, the information flow between IPs is also an important part in any SoC environment. The 3PIPs play an important role in data computation acceleration and some security-critical functions, e.g., encryption and hashing.

The authors in [20] extended the bus to support extra tag transmission, although this DIFT design may not be deployed in real-world systems. A possible solution is to attach one DIFT-oriented IP to the bus. The IP is responsible for monitoring the information flow between IPs and provide a summary to the DIFT engine in the processor or coprocessor for further processing. In addition, an extra bus topology can be added to support the tag transmission at runtime. The extra bus topology can be used for propagating the tag about information flow between processor and IP. The tag propagation between IPs can also leverage this new bus topology. The DIFT engine in the processor and coprocessor can serve as a management unit for information flow in the whole SoC.

### 6.6. Future Research

Software and hardware co-design is a good future direction. This type of design can combine the benefits of both hardware-based design and software-based design. Further, off-core-based design can package all DIFT operations into one dedicated hardware component and introduce few modifications to the processor core. The target software is instrumented through binary instrumentation mechanisms or compiler aided tools. The inserted auxiliary instructions are utilized to instruct the dedicated hardware component to accomplish tag propagation and checking. In addition, the problem of implicit flow can be processed at the software side.

The common tag designs include single-bit, multi-bit and policy oriented. In order to enforce security policy in terms of combinations of tags, the granularity of tags should be defined according to the types of system resources, e.g., process, even, file and variable. Additionally, the tag should record the lifetime and the specific activity of the corresponding resource. It can serve as a log and enable the programmer or user to replay the action on specified resource. Therefore, the possible security risks can be analyzed.

Along with protecting the target software at runtime, the security of SoC should also be considered. Modern SoC includes a lot of third-party IPs which may compromise the security of the whole system. Tracking the information flow on the whole SoC can help the system designer define suitable security policy. For the bus, the tag can be transmitted in parallel with the shake signal between IPs. Additionally, the bus can be extended to support the tag transmission. For security checking, the corresponding operations can be packaged into one dedicated hardware component or accomplished in IP wrapper.

## 7. Conclusions

In this paper, we analyzed a large number of DIFT designs and reviewed all those designs in the context of three categories, software DIFT, hardware DIFT and hardware-software co-design DIFT. We discussed the advantages and disadvantages of each DIFT framework with focuses on the performance overheads and security protection levels. Based on our discussion, we presented new research directions in this area. We hope that this survey will help researchers in related areas to develop more efficient DIFT frameworks to help track information flow within the computing systems.

## Figures and Tables

**Figure 1 micromachines-12-00898-f001:**
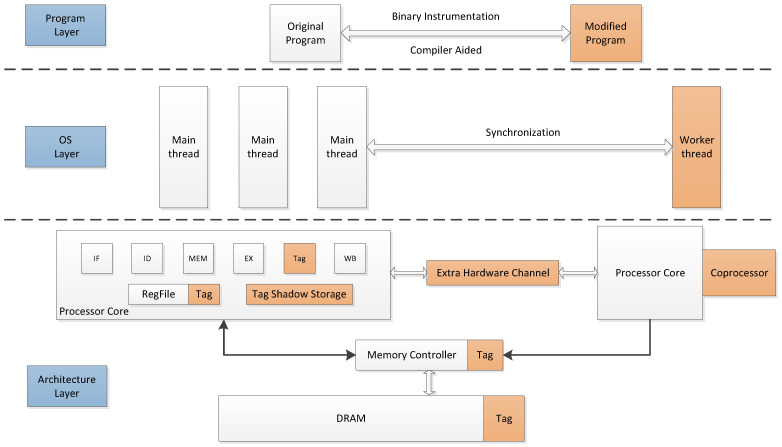
The DIFT works on different layers of whole systems.

**Table 1 micromachines-12-00898-t001:** Details of dynamic information flow tracking techniques. CFI, DFI and NI are abbreviations of control-flow integrity, data-flow integrity and new infrastructure, respectively.

#	Tag	Type	CFI	DFI	NI	Performance	Hardware
[7]	1 bit/byte	SW-Program level	×	✓	×	650%	-
[12]	1 byte, 1 bit/byte	SW-Program level	×	×	×	224–700%	-
[10]	variable tag	SW-Program level	×	×	✓	3000–5000%	-
[11]	32 bit/variable	SW-Program level	×	✓	×	14%	-
[13]	Process level	SW-Program Level	×	×	✓	5.35%	-
[1]	kernel object label	SW-OS Level	×	×	✓	Unspecified	-
[6]	object label	SW-System Level	×	✓	✓	5600%	-
[2]	2 bit/byte	SW-Offloading	×	×	✓	48%	-
[15]	gate logic	HW-Gate Level	×	✓	×	Unspecified	70% ALUT
[21]	4 bit/word	HW-In Core	✓	✓	×	234%	22% BRAM, 42% LUT
[22]	1 bit/byte	HW-In core	×	✓	×	Negligible	12.5% BRAM, 0.82% LUTs, ≤1% Area
[23]	4 bit/register	HW-In core	✓	×	×	-	-
[27]	1 bit/word	HW-Off Core	×	×	✓	1–3.7%	Unspecified
[28]	4 bit/word	HW-Off Core	×	×	✓	0.79%	16% BRAM, 7.64% LUTs
[24,25]	1 bit/word	HW-Off core	✓	✓	×	1.6%	60% BRAM, 28.36% LUTs
[34]	1 bit/tens of bytes	HW-Off core	✓	✓	×	50–60%	4–5% Memory, 5% Power
[32]	1 bit/byte	HW-Inter Core	×	×	✓	9.7–12%	Unspecified
[19]	bus request tag	HW-IP wrapper	×	✓	×	Unspecified	31% Area
[20]	variable tag size	HW-SoC DIFT	×	×	✓	7.25%	Unspecified
[31]	Extra tag register	HW-SW Codesign	×	✓	×	0–100%	Unspecified
[41]	1 bit/word	HW-SW Codesign	✓	×	×	Unspecified	3.125% Memory
[33]	pointer-sized tag/word	HW-SW Codesign	✓	✓	✓	10–40%	Unspecified
[29]	single bit	HW-Tagged Memory	×	✓	✓	5%	15.4% logic elements
[30]	2 bit	HW-Tagged Memory	✓	✓	✓	25.2%	6.25% Memory

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
