# Peer review of "Dynamic Information Flow Tracking: Taxonomy, Challenges, and Opportunities"

_micromachines, 2021, doi:10.3390/mi12080898_

Round 1
Reviewer 1 Report
Dynamic information flow tracking (DIFT) is crucial for the security of modern computing systems. The authors provide a comprehensive description, categorizing the designs into software-based, hardware-based, and co-design-based. Under each category, further classifications and corresponding descriptions are clearly delivered, making the paper easy to follow. Overall, this is a well-written introduction to DIFT, and the reviewer enjoyed reading it.
While as a paper for the research community, more literature from recent years may be desired. The reviewer will suggest including one or two recent pieces of literature (e.g., after 2017) in each sub-section, if not yet. The comments on the limitations of current techniques are sufficient. More thoughts on possible solutions will make this paper even more insightful.
Author Response
Response1.1
Thank you very much for the constructive reviews regarding our manuscript Dynamic Information Flow Tracking: Taxonomy, Challenges, and Opportunities. We added and discussed the latest references on different types of DIFT design. In addition, a new type of hardware based DIFT design, called Tagged Memory Architecture, has been added. The main goal of this design is to protect memory at runtime through information flow tracking, so that the whole memory system is extended to support tag-related operations. We have carefully revised the whole paper and added some latest references are presented below.
3.2.2. In-Core DIFT Design
Different to above in-core based design, Weiwei et al. in [23] simplify the common DIFT framework to only track registers. The states of registers will be analyzed further to determine whether the Return Oriented Programming (ROP) attacks happened. In order to reduce the modification on existing processor architecture, the taint related information is stored in the shadow renaming tables and reorder buffer.
3.3. Tagged Memory Architecture
Robert et al. in [29] propose a complete tagged memory architecture to support information flow tracking and security analysis. A single-bit tag shadow space is used to record the metadata at runtime. An in-DRAM tag table and tag Cache are utilized to manage the tag in whole system. Samuel et al. in [30] present a new tagged memory architecture to combine compartmentalization (e.g., Intel SGX [42]) and isolated execution environments (e.g., TrustZone [43]). Two extra tag bits are utilized to differentiate privilege levels and security domains. A dedicated hardware serves as the trust root to enforce security policies in terms of tag and finish security domains switch. Also, extra tag-aware instructions are added to support flexible tag management.
4.1. System Level Dynamic Information Flow Tracking
Meisam et al. in [6] propose a whole system level DIFT framework to resist in-memory injection attacks which are difficult to be detected using current malware analysis solutions. In order to capture the indirect flow in security applications, they introduce different types of tag instead of expressing different security policies in terms of similar tag structure. Besides, tag conveys rich provenance information, e.g., lifetime of data byte and activities associated with the data byte. Further, indirect flows can be processed in different security policies according to the tag type. Similar with other system level DIFT solutions, significant performance and memory overheads incurred(56x performance slowdown is introduced on QEMU based simulator platform).
4.2. Program Level Dynamic Information Flow Tracking
Sanchuan et al. in [8] present an efficient static information flow tracking method in order to reduce performance overhead. Instead of using DBI based methods, they adopt static binary instrumentation (SBI) to scan sources of taint and select the instructions need to be instrumented and monitored. They propose an approach to statically identify the instructions will involve tainted memory or registers through value set analysis (VSA).
Yang et al. in [13] use record-replay technology to develop a log based system to record the system call events and related activities. Based on the log collected at runtime, a provenance graph is constructed to improve the system performance. The unrelated instructions and processes are filter out using the reachability analysis on the provenance graph. However, the memory overhead introduced is 50%.
In addition to software based DIFT design, Andrew et al. in [14] use verify practical security architecture through static information flow analysis. The target security architecture can be verified with information flow control (IFC) at the level of the hardware description language (HDL). The main goal of IFC HDLs is to guarantee the security property noninterference at the design stage.
Response1.2
In Section 6.6, we added the analysis about future research on DIFT techniques from three perspectives - the system design, the tag design, and SoC protection. The software and hardware co-design framework is an exceptional solution. As to the hardware part, the DIFT related operations can be combined into one dedicated hardware component, e.g., coprocessor in off-core design. At the same time, the target software needs to be instrumented to instruct the dedicated hardware component to perform different DIFT operations. Further, the tag-based protection mechanisms can be utilized to protect sensitive information on the SoC level, e.g., the sensitive information on IP. In light of this discussion, we have added an extra section contain the following text.
6.6. Future Research
The software and hardware co-design is a good future direction. This type of design can combine both the benefits of hardware based design and software based design. Further, off-core based design can package all DIFT operations into one dedicated hardware component and introduce little modification on the processor core. The target software is instrumented through binary instrumentation mechanisms or compiler aided tools. The inserted auxiliary instructions are utilized to instruct the dedicated hardware component to accomplish tag propagation and checking. In addition, the problem of implicit flow can be processed at the software side.
The common tag design includes single-bit, multi-bit or policy oriented. In order to enforce security policy in terms of combination of tag, the granularity of tag should be defined according to the type of system resource, e.g., process, even, file and variable. Also, the tag should record the lifetime and the specific activity of the corresponding resource. It can serve as a log and enable the programmer or user to replay the action on specified resource. Therefore, the possible security risks can be analyzed.
Instead of protecting the target software at runtime, the security of SoC should also be considered. Modern SoC includes a lot of third-party IPs which may compromise the security of whole system. Tracking the information flow on the whole SoC can help the system designer define suitable security policy. For bus, the tag can be transmitted in parallel with the shake signal between IPs. Also, the bus can be extended to support the tag transmission. For security checking, the corresponding operations can be packaged into one dedicated hardware component or accomplished in IP wrapper.
Reviewer 2 Report
This paper proposes a survey on the current Dynamic Information Flow Tracking (DIFT) solutions and analyzes the features and limitations of these solutions.
Here are some comments which can help to improve the paper.
1) I cannot even see a single citation in the Introduction section, while in survey paper, it is expected a comprehensive citation strategy. I suggest authors to revise the Introduction and provide enough citations in the text.
2) I would expected to see something about other survey papers in the Introduction. Is there any other survey article for DIFT? What are the problems in the previous survey papers?
3) Most of the provided references are outdated and I suggest to address this by reviewing some recently published articles.
Author Response
Response2.1
Thanks for your comments, and your suggestion is really helpful.
We added the discussions about the existing DIFT-related surveys and then analyzed the limitations of these works in the Introduction section. We have carefully revised the whole paper and added some discussions are presented below.
Also, there are some surveys about existing DIFT works. Fabian et al. in [36] summarize and analyze several specified DIFT works. Ori et al. in [37] aim at analyzing the existing methods of malware detection and analysis. Also, this work does not involve static methods. Ping et al. in [38] mainly focus on the DIFT works on mobile malware detection. They did not give a comprehensive analysis on existing DIFT works on different application environments. Lok et al. in [39] analyze the precision and soundness sof existing software based DIFT works. Daniele et al. in [40] analyze existing Dynamic Binary Instrumentation (DBI) methods. Also, they discuss the usage of DBI on DIFT framework.
Response2.2
We added and discussed the latest references on different types of DIFT design. In addition, a new type of hardware based DIFT design, called Tagged Memory Architecture, has been added. The main goal of this design is to protect memory at runtime through information flow tracking, so that the whole memory system is extended to support tag-related operations. In light of this discussion, we have added some latest references contain the following text.
3.2.2. In-Core DIFT Design
Different to above in-core based design, Weiwei et al. in [23] simplify the common DIFT framework to only track registers. The states of registers will be analyzed further to determine whether the Return Oriented Programming (ROP) attacks happened. In order to reduce the modification on existing processor architecture, the taint related information is stored in the shadow renaming tables and reorder buffer.
3.3. Tagged Memory Architecture
Robert et al. in [29] propose a complete tagged memory architecture to support information flow tracking and security analysis. A single-bit tag shadow space is used to record the metadata at runtime. An in-DRAM tag table and tag Cache are utilized to manage the tag in whole system. Samuel et al. in [30] present a new tagged memory architecture to combine compartmentalization (e.g., Intel SGX [42]) and isolated execution environments (e.g., TrustZone [43]). Two extra tag bits are utilized to differentiate privilege levels and security domains. A dedicated hardware serves as the trust root to enforce security policies in terms of tag and finish security domains switch. Also, extra tag-aware instructions are added to support flexible tag management.
4.1. System Level Dynamic Information Flow Tracking
Meisam et al. in [6] propose a whole system level DIFT framework to resist in-memory injection attacks which are difficult to be detected using current malware analysis solutions. In order to capture the indirect flow in security applications, they introduce different types of tag instead of expressing different security policies in terms of similar tag structure. Besides, tag conveys rich provenance information, e.g., lifetime of data byte and activities associated with the data byte. Further, indirect flows can be processed in different security policies according to the tag type. Similar with other system level DIFT solutions, significant performance and memory overheads incurred(56x performance slowdown is introduced on QEMU based simulator platform).
4.2. Program Level Dynamic Information Flow Tracking
Sanchuan et al. in [8] present an efficient static information flow tracking method in order to reduce performance overhead. Instead of using DBI based methods, they adopt static binary instrumentation (SBI) to scan sources of taint and select the instructions need to be instrumented and monitored. They propose an approach to statically identify the instructions will involve tainted memory or registers through value set analysis (VSA).
Yang et al. in [13] use record-replay technology to develop a log based system to record the system call events and related activities. Based on the log collected at runtime, a provenance graph is constructed to improve the system performance. The unrelated instructions and processes are filter out using the reachability analysis on the provenance graph. However, the memory overhead introduced is 50%.
In addition to software based DIFT design, Andrew et al. in [14] use verify practical security architecture through static information flow analysis. The target security architecture can be verified with information flow control (IFC) at the level of the hardware description language (HDL). The main goal of IFC HDLs is to guarantee the security property noninterference at the design stage.